# The Impact of Clinical and Histopathological Factors on Disease Progression and Survival in Thick Cutaneous Melanomas

**DOI:** 10.3390/biomedicines11102616

**Published:** 2023-09-23

**Authors:** Dana Antonia Țăpoi, Diana Derewicz, Ancuța-Augustina Gheorghișan-Gălățeanu, Adrian Vasile Dumitru, Ana Maria Ciongariu, Mariana Costache

**Affiliations:** 1Department of Pathology, Carol Davila University of Medicine and Pharmacy, 020021 Bucharest, Romania; dana-antonia.tapoi@drd.umfcd.ro (D.A.Ț.); ana-maria.ciongariu@drd.umfcd.ro (A.M.C.); mariana.costache@umfcd.ro (M.C.); 2Department of Pathology, University Emergency Hospital, 050098 Bucharest, Romania; 3Department of Pediatrics, Carol Davila University of Medicine and Pharmacy, 020021 Bucharest, Romania; diana.derewicz@umfcd.ro; 4Department of Pediatric Hematology and Oncology, Marie Sklodowska Curie Clinical Emergency Hospital, 041447 Bucharest, Romania; 5Department of Cellular and Molecular Biology and Histology, Carol Davila University of Medicine and Pharmacy, 020021 Bucharest, Romania; ancuta.gheorghisan@umfcd.ro; 6C.I. Parhon National Institute of Endocrinology, 011863 Bucharest, Romania

**Keywords:** thick cutaneous melanoma, Breslow, necrosis, perineural invasion, prognosis, survival

## Abstract

Thick cutaneous melanomas (Breslow depth > 4 mm) are locally advanced tumors, generally associated with poor prognosis. Nevertheless, these tumors sometimes display unpredictable behavior. This study aims to analyze clinical and histopathological features that can influence the prognosis of thick melanomas. This is a retrospective study on 94 thick primary cutaneous melanomas diagnosed between 2012 and 2018 that were followed-up for at least five years to assess disease progression and survival. We evaluated the age, gender, tumor location, histological subtype, Breslow depth, Clark level, resection margins, mitotic index, the presence/absence of ulceration, necrosis, regression, microsatellites, neurotropism, lymphovascular invasion, and the pattern of tumor-infiltrating lymphocytes, and their association with disease progression and survival. By conducting univariate analysis, we found that progression-free survival (PFS) was significantly associated with female gender, the superficial spreading melanoma (SSM) subtype, mitotic index, necrosis, microsatellites, and perineural invasion. Overall survival (OS) was significantly associated with female gender, Breslow depth, SSM subtype, necrosis, microsatellites, and perineural invasion. Through multivariate Cox proportional hazards regression, we found that the only factors associated with PFS were Breslow depth, necrosis, microsatellites, and perineural invasion, while the factors associated with OS were Breslow depth, necrosis, microsatellites, and perineural invasion. Certain histopathological features such as Breslow depth, necrosis, microsatellites, and perineural invasion could explain differences in disease evolution. This is one of the first studies to demonstrate an association between necrosis and perineural invasion and outcomes in patients with thick melanomas. By identifying high-risk patients, personalized therapy can be provided for improved prognosis.

## 1. Introduction

Cutaneous melanomas represent aggressive malignancies, often diagnosed in locally advanced stages. They are responsible for 75% of all skin cancer deaths, even though they account for only 4% of all skin cancer cases [1].

Melanoma is a heterogeneous disease with widely variable clinical, histopathological, immunohistochemical, and molecular features that influence treatment and prognosis. Wallace Clark was the first to classify melanomas based on histological traits, thus describing lentigo maligna melanoma (LMM), superficial spreading melanoma (SSM), and nodular melanoma (NM) [2,3]. In addition, Clark also envisioned a system for classifying cutaneous melanomas based on the depth of invasion by anatomic sites [2,3], which can be correlated with the outcome of patients [4]. Later, Alexander Breslow developed a different classification based on the depth of invasion in millimeters [5]. Even though the initial cut-off values for each Breslow stage have changed, the Breslow classification is still one of the best predictors of survival and must be reported, according to the latest American Joint Committee on Cancer (AJCC) staging guide [1,6].

Apart from the Breslow level of invasion, several other histological features are considered of adverse prognostic significance: the presence of microsatellites [7], lymphovascular invasion [7,8], high mitotic rates, non-brisk tumor-infiltrating lymphocytes (TILs) pattern, nodular melanoma subtype, ulceration, and neurotropism [9,10,11,12,13]. The presence of regression, on the other hand, is a controversial prognostic factor associated with both positive and negative outcomes [14,15,16].

Finally, clinical features such as age, gender, and tumor location seem to influence the prognosis. In this respect, male gender and older age have been associated with a poorer prognosis by some authors [17,18,19], while others found no such correlation [20,21]. Considering the anatomic site of the primary cutaneous melanomas, some authors demonstrated a worse prognosis for palm/sole localization [19], and others for the head and neck [22,23].

Nevertheless, the impact of these prognostic markers is not well established for thick melanomas (>4 mm) as various authors reported inconsistent results concerning their predictive value in thick melanomas [24,25,26,27]. Therefore, this study aims to analyze clinical and histological features for evaluating the progression-free survival (PFS) and overall survival (OS) in primary cutaneous melanomas > 4 mm.

## 2. Materials and Methods

### 2.1. Study Participants

This retrospective study included 94 patients diagnosed with primary cutaneous melanomas at the University Emergency Hospital of Bucharest, Romania, between 2012 and 2018. The cases were collected by performing a retrospective chart review. Initially, 111 patients with thick cutaneous melanomas were identified, but 17 were excluded due to lack of follow-up. The inclusion criteria for the patients were:A histopathological diagnosis of thick cutaneous melanoma (>4 mm);A whole-body CT scan after the initial diagnosis to assess the presence of metastases;Follow-up of five years to assess PFS and OS. PFS was calculated as the time between the date of the primary diagnosis and metastatic spread, and OS was calculated as the time between the date of the initial diagnosis and death.

This study was approved by the Ethics Committee of the University Emergency Hospital of Bucharest, Romania, and was conducted in accordance with the principles of the Helsinki Declaration. Each patient signed an informed consent form.

### 2.2. Histopathological and Immunohistochemical Analysis

The tissue samples were processed by standard histopathological methods. Two pathologists (D.A.Ț. and A.V.D.) examined the sections to establish the diagnosis. Differences in opinion were settled by consultation with a third pathologist (M.C.).

### 2.3. Data Collection and Analysis

The following variables were collected for each patient: age, gender, primary tumor localization, histological subtype of the melanoma, Breslow depth, Clark level, resection margins, mitotic index, the presence/absence of ulceration, necrosis, regression, microsatellites, neurotropism, lymphovascular invasion, the pattern of TILs, and the presence/absence of metastatic lesions and the time until disease progression or death. All the cases were revised before performing statistical analyses, and melanoma subtype was classified according to the fifth edition of the WHO Classification of Tumours Editorial Board—skin tumours [28]. TMN staging was assessed using the eighth edition of the cancer staging manual of the American Joint Committee on Cancer (AJCC) [29]. Descriptive statistics were provided, including mean, standard deviation (SD), median, and range for continuous variables and frequency with percentage. Univariate and multivariate Cox proportional hazards regression analyses were performed on all variables with PFS and OS as the outcome. We reported hazard ratios (HR) with 95% confidence intervals (CI) and *p* values, considered significant at *p* < 0.5. The Kaplan–Meier product limited method was used to estimate survival probabilities, and log-rank test comparisons were made. Statistical analysis was performed using GraphPad Prism 9.0 (Graphpad Software Inc., San Diego, CA, USA).

## 3. Results

### 3.1. Demographic and Clinical Characteristics of the Study Population

The mean age at diagnosis of the entire group was 65.01 years old (standard deviation 14.06; range 24–89); 45.75% (n = 43) were women and 54.25% (n = 51) were men. The mean age of the female patients was 64.16 (SD 14.35), and the mean age of the male patients was 64.88 (SD 13.95). The difference was not significant (*p* = 0.9239, unpaired *t*-test). During follow-up, 56 patients developed metastases, and 51 died. The age of the patients with PFS was slightly younger (mean 63.45; SD 16.11) than that of the patients with progressive disease (mean 66.07; SD 12.52). The age at diagnosis of patients who survived during follow-up was also slightly younger (mean 64.05; SD 15.67) than those who died during follow-up (mean 65.82; SD 12.82; range 35–89).

Among the patients with PFS, 60.52% were women (n = 23), and 39.48% were men (n = 15). In the metastases group, 35.71% (n = 20) were women and 64.29% (n = 36) were men. Among the patients who died during follow-up, 39.53% (n = 17) were female and 66.67% (n = 34) were male.

Regarding localization of the tumors, the most affected region was the trunk, followed by the limbs (Figure 1).

The clinical characteristics of PFS and OS are presented in Table 1.

### 3.2. Histopathological Characteristics of the Study Population

#### 3.2.1. The Depth of Invasion

The first histological feature we analyzed was the depth of invasion, as our study included only thick melanomas (Breslow > 4 mm). Therefore, the mean Breslow depth in the study population was 11.05 mm (median 8.65; range 4.1–46). The median Breslow depth in patients without metastases was 5.95 (range 4.1–17), and the median Breslow depth in patients with metastases was 10.4 (range 4.2–46) (Figure 2).

The median Breslow depth in patients who survived during follow-up was 6 (range 4.1–17), and the median Breslow depth between patients who died was 11.8 (range 4.2–46) (Figure 3).

We also evaluated Clark’s level of invasion. As we included only thick melanomas, all the cases had at least invasion in the reticular dermis (Clark level IV) or the subcutis (Clark level V). Clark level IV was noted in 38.89% (n = 14) of the cases without progressive disease and 39.29% (n = 22) of the patients with progressive disease. Additionally, Clark level IV was observed in 37.21% (n = 16) of the patients who survived and in 39.22% (n = 20) of the people who died. The relationship between the depth of invasion and PFS and OS is presented in Table 2.

#### 3.2.2. Melanoma Subtype

The most frequently encountered histological subtype was nodular melanoma (NM), followed by superficial spreading melanoma (SSM), while the least encountered was lentigo maligna melanoma (LMM). The melanoma subtypes encountered in our study are presented in Figure 4.

NM was the most diagnosed subtype in patients with progressive disease, followed by acral lentiginous melanoma (ALM). In patients who died, the most encountered subtype was also NM, followed by ALM, while none of the patients with LMM died of disease during follow-up (Table 3).

By conducting Cox univariate analysis, we found that SSM was significantly associated with both PFS (HR = 0.37, 95%CI: 0.14–0.81, *p* = 0.0241) and OS (HR = 0.35, 95%CI: 0.14–0.8, *p* = 0.025).

#### 3.2.3. Mitotic Index

The mean mitotic index in the whole group was 8.9 (median 8; range 2–23). The mean mitotic index was 9.63 in the metastatic group (median 8; range 2–21) and 7.81 in patients without progressive disease (median 5.95; range 2–23) (Figure 5).

The mean mitotic count in patients who survived during follow-up was 8.047 (median 7; range 2–23), and for patients who died, the mean mitotic count was 9.627 (median 8; range 2–21) (Figure 6).

By conducting Cox univariate analysis, we found that the mitotic count was associated with PFS (HR = 1.05, 95%CI: 1.001–1.101, *p* = 0.0349) but not with OS (HR = 1.05, 95%CI: 0.99–1.01, *p* = 0.06090).

#### 3.2.4. Ulceration, Necrosis, Regression, Microsatellites, Lymph-Vascular Invasion and Neurotropism

We evaluated the presence of the following features: ulceration, necrosis, regression, micro-satellites, lymphovascular invasion and neurotropism in relation to PFS (Table 4).

Even though the presence of ulceration and lymphovascular invasion were more frequently encountered in the metastatic group, and the presence of regression was more frequent in patients without metastases, none of these findings were statistically significant. Necrosis, microsatellites, and neurotropism, however, correlate strongly with metastatic disease.

We also analyzed the presence of the aforementioned features in relation to OS (Table 5), and the results were similar. The presence of ulceration and lymphovascular invasion was more frequent in patients who died during follow-up, and the presence of regression was more frequent in patients who survived, but the results were not significant. Necrosis, microsatellites and neurotropism are associated with decreased OS.

#### 3.2.5. Tumor-Infiltrating Lymphocytes (TILs)

We analyzed the TILs to determine the prognostic difference between brisk vs. non-brisk patterns. Brisk inflammatory infiltrate was noted in 18.42% (n = 7) of the cases without metastases and 3.57% (n = 2) patients with metastases. The difference was not statistically significant (HR = 0.26, 95%CI: 0.04–0.84, *p* = 0.0615). Brisk inflammatory infiltrate was present in 16.28% (n = 7) of the patients who survived and 3.92% (n = 2) of the patients who died. The pattern of TILs also failed to predict OS (HR = 0.31, 95%CI: 0.05–0.98, *p* = 0.1001).

#### 3.2.6. Resection Margins

We analyzed the difference between the resection margin closest to the tumors. The mean value in the study population was 7.055 mm (median 5.15; range 0–28). The mean value for patients without metastases was 7.943 mm (median 5.7, range 0–24), and the mean value for patients with metastases was 6.452 mm (median 5.05; range 0–28), but the difference was not significant (HR = 0.97, 95%CI: 0.92–1.006, *p* = 0.1106). The mean value for patients who survived during follow-up was 7.42 mm (median 5.2; range 0–24), and for patients who died, it was 6.74 mm (median 5.1; range 0–28). Therefore, resection margins cannot be correlated with OS (HR = 0.97, 95%CI: 0.93–1.02, *p* = 0.231).

### 3.3. Multivariate Analysis for PFS and OS in Thick Melanomas

Multivariate analysis of the factors associated with PFS revealed that only the presence of necrosis, microsatellites, neurotropism, and Breslow depth are independently associated with progressive disease (Table 6).

Multivariate analysis for predicting OS showed that necrosis, microsatellites, neurotropism, and Breslow depth are independently associated with decreased OS (Table 7).

### 3.4. Survival Prediction

The final part of our study was focused on analyzing survival rates for patients with thick melanomas based on the Breslow depth of invasion having the following cut-off values: tumors < 8 mm, tumors between 8 and 10 mm, tumors between 10 and 15 mm, and tumors >15 mm (Figure 7).

The patients were followed up for an average of 50.04 months (median 58, range 6–120). The group with Breslow < 6 mm encompassed 39 patients, out of which 29.03% (n = 9) died; the group with Breslow between 6 and 8 mm included 12 patients, out of which 41.66% (n = 5) died; the group with Breslow between 8 and 10 mm included 16 patients, out of which 62.5% (n = 10) died; the group with Breslow between 10 and 15 mm included 17 patients, out of which 64.7 (n = 11) died; and the group with Breslow >15 mm encompassed 18 patients, out of which 88.88% (n = 16) died (*p* < 0.0001). Additionally, for patients who died during follow-up, the median survival time decreased as the Breslow depth increased from 40 months in patients with Breslow < 6 mm to 17 months for patients with Breslow >15 mm, but the difference was not significant (*p* = 0.0872, Kruskal–Wallis test).

## 4. Discussion

Cutaneous melanomas with Breslow > 4 mm are associated with the worst survival rates [30,31]. Nevertheless, not all thick melanomas behave the same way, and little is known about additional prognostic factors for such tumors. In our study population, the age of the patients was not correlated with progressive disease, and similar findings were noted by other authors [26,32,33]. In contrast, others demonstrated that increased age is an adverse prognostic factor in thick melanomas [34,35,36,37]. Univariate analysis revealed that female patients have significantly better PFS and OS than male patients, but multivariate analysis found no association between gender and prognosis. Data regarding this relationship in thick melanomas are inconsistent. Some authors found no association between gender and OS in thick melanomas [26,35,36]. El Sharouni M.A. et al. demonstrated in a multivariate analysis that the male gender is associated with decreased OS [34], and Han D. et al. showed that the male gender is associated with decreased OS only in thick melanomas with a Breslow depth <10 mm [27].

The location of the primary tumor did not influence PFS or OS in our population. These findings are similar to those presented by multiple studies [27,33,34,35,36,38]. On the contrary, Blakely A. et al. noted that head and neck tumors are independently associated with decreased PFS but not OS [26]. Kachare S.D. et al. showed that location on the trunk was independently associated with decreased OS [39].

Melanoma subtype is another parameter that should be reported for thick melanomas. From the univariate analysis, we found that SSM was a significant predictor for increased PFS and OS, while NM and ALM are associated with a worse prognosis. It is interesting that no patient with LMM died, but the number of cases (n = 2) is too small to draw reliable conclusions for this subtype. By undertaking multivariate analysis, we discovered that the melanoma subtype was not an important predictor for PFS or OS. These findings are interesting as NM and ALM melanomas are generally associated with decreased survival and are an adverse prognostic factor for thick melanomas as well, being associated with sentinel lymph node metastases [25,38]. Nevertheless, the exact association between histological subtype and OS in thick melanomas remains to be determined, as some other studies found no correlation between the two [32,33].

The Breslow depth of invasion was a significant predictor for both PFS and OS in univariate and multivariate analyses. Most authors confirm that Breslow depth is still an important prognostic factor in thick melanomas [26,32,35,38,40,41,42]. Even though increasing tumor thickness remains an adverse prognostic factor, this relationship may lose its prognostic significance in ultra-thick melanomas as some studies note that ultra-thick melanomas with Breslow >10 mm [27] or Breslow >15 mm [24] are not associated with decreased PFS or OS, in comparison to thick melanomas. In our study group, however, survival significantly decreased as Breslow depth increased. Clark’s level of invasion had no prognostic value in our study. These results were expected as all the tumors included had at least Clark level IV, but our study is one of the very few to assess this parameter in thick cutaneous melanomas. A 2014 study provided similar findings, with no association between Clark level and PFS or OS [40].

Higher mitotic count was associated with decreased PFS but not with reduced OS during univariate analyses. However, these findings were not confirmed by multivariate analyses. These results concord with some studies about thick cutaneous melanomas [26,41,43,44]. On the contrary, other authors demonstrated that mitotic rate is associated with progressive disease and decreased OS. Still, these studies analyzed mitotic figures as <1/mm^2^ vs. ≥1/mm^2^ [40] or as present vs. absent [32] rather than as a continuous variable. Yamamoto M. et al. demonstrated that mitotic count is an independent adverse prognostic factor using the cut-off of 5 mitoses/mm^2^ [45]. In our study, the minimum mitotic count was 2/mm^2^, and we believe that since thick melanomas are advanced tumors with high mitotic figures, this parameter should be analyzed as a continuous variable.

The presence of ulceration is mandatory to be reported as it influences tumor staging, according to the AJCC Cancer Staging Manual [29]. For thick melanomas, some authors demonstrated that ulceration is associated with decreased PFS and OS [34,35,39,40,42]. Most tumors in our study were ulcerated, and this feature was not associated with PFS or OS. Similar results were noted by Blakely A. et al. [26], Ruskin O. et al. [43], Yamamoto M. et al. [45], and Gyorki D.E. et al. [46], while Rodriguez Otero J.C. et al. [33] and Bello D. et al. [36] found ulceration to be a significant adverse prognostic factor only in univariate analysis. Interestingly, Han D. et al. found a correlation between ulceration and PFS and OS only for thick melanomas with a Breslow depth between 4 and 6 mm [27]. Additionally, the same authors found ulceration predictive for sentinel lymph node metastases but not for OS [38].

The presence of necrosis was strongly associated with decreased PFS and OS in univariate analyses and remained an adverse prognostic factor for PFS and OS in multivariate analysis. Our research is one of the few studies analyzing the impact of necrosis in thick melanomas. Ladstein R.G. et al. demonstrated that necrosis is independently correlated with decreased OS but did not analyze this feature in relationship with PFS [44].

The presence of regression is still a debatable prognostic factor for cutaneous melanoma in general [14,15,16] and is scarcely analyzed in thick melanomas. In our study, regression was more frequently noted in cases with increased PFS and OS, but it was not statistically significant. Song Y. et al. [41] and Gyorki D.E. et al. [46] also reported a lack of association between regression and prognosis in thick melanomas. At the same time, Cintolo J.A. demonstrated that the presence of regression correlated with decreased survival [47]. However, Ribero S. et al. reported that regression is independently associated with increased PFS and OS [42].

In our study, microsatellites were a strong independent predictor for decreased PFS and OS. Similar results were reported by Gyorki D.E. et al. [46], while Yamamoto M. et al. found an association between microsatellites and decreased PFS only in univariate analysis [45]. These differences might be explained by the fact that microsatellites may be a prognostic factor dependent on Breslow depth. Han D. et al. demonstrated that microsatellites are an independent factor for decreased OS only in thick melanomas with thicknesses between 4 and 8 mm [38] and for PFS in tumors with thicknesses between 4 and 6 mm [27].

Lymphovascular invasion is considered a strong independent predictor for decreased PFS and OS in melanomas in general [48] but reports on its association with thick melanomas vary. Song Y. et al. demonstrated that lymphovascular invasion remains an independent predictor for OS in thick melanomas [41]. At the same time, Han D. et al. noted that lymphovascular invasion is prognostic for PFS and OS only in melanomas with a Breslow depth between 4 and 6 mm [27]. On the contrary, we found no correlation between lymphovascular invasion and PFS or OS in thick melanomas, and numerous studies confirm these observations [26,36,37,43,46].

According to the AJCC staging guide, it is recommended to report the presence of neurotropism as this feature is associated with an increased risk of local recurrence [29]. Consequently, some melanoma centers treat such patients with postoperative radiotherapy [49]. However, little is known about the impact of perineural invasion on long-term prognosis. Namikawa K. et al. found a significant association between perineural invasion and decreased PFS and OS in univariate analyses but not multivariate analyses [50]. Similarly, Vița O. et al. demonstrated that perineural invasion is associated with in-transit metastases [51]. Furthermore, to the best of our knowledge, the impact of neurotropism on prognosis in thick melanomas has not been thoroughly assessed until now. Interestingly, in our study, perineural invasion was a strong independent prognostic factor for both PFS and OS. This may represent a particular feature of thick melanomas, but more studies are needed to confirm it.

The presence of TILs (brisk and non-brisk) was associated with a positive prognosis in thick melanomas [47], while other authors found no such association [46]. In our study, all the cases had at least focal intratumoral lymphocytes, and we evaluated where the pattern of TILs (brisk vs. non-brisk) influences prognosis. Even though brisk inflammatory infiltrate was more often observed in cases without metastases, our study found no correlation between the TILs pattern and PFS or OS. Several other authors reported no association between TILs and prognosis [36,37].

In our study, the resection margins, as a continuous variable, did not predict PFS or OS. Nevertheless, patients without progressive disease and patients who survived had resection margins around 1 mm larger on average than patients with progressive disease or patients who died. This small difference may not be enough to significantly improve the prognosis of patients with cutaneous melanoma. A 2013 study demonstrated that resection margins, as a continuous variable, are independently associated with local and locoregional disease-free survival but not OS [52]. On the contrary, Blakely A.M. analyzed resection margins as categorial variables (≥2 cm vs. <2 cm) but did not correlate this feature with PFS or OS [26]. Additionally, a study on head and neck thick melanomas analyzing reaction margins as categorial variables (0–1 cm, 1–2 cm, and 2–3.5 cm) found no association between this parameter and PFS or OS [43].

Dermatoscopic features may also offer important clues for disease severity. Cutaneous melanomas can mimic a great variety of other lesions, both benign and malignant [53]. Nevertheless, in cases of thicker melanomas, there are some dermatoscopic features more frequently encountered in comparison to other skin tumors, such as marked asymmetry of color and structure, the presence of a blue and black color on the surface, or ulceration [54]. These findings reflect our study population. The dermatoscopic features were not statistically analyzed in this study, as most cases, being locally advanced melanomas, were clinically suggestive of a melanocytic malignancy. Nevertheless, dermatoscopy remains an important diagnostic tool for thinner melanomas.

As the patients included in this study presented with advanced tumors, they were treated with adjuvant therapy and/or therapy for metastatic disease. The therapies included most often interferon α and dacarbazine as these were the main therapeutic options available at the beginning of our study. Immune-checkpoint inhibitors first became available for metastatic disease in 2017 and for patients with stage III disease in 2019, while BRAF inhibitors first became available for systemic disease in 2015 and for patients with stage III disease in 2020. Additionally, in Romania, none of these drugs are currently approved for thick cutaneous melanomas without lymph node metastases. All these aspects may be a reason for the low PFS and OS in our study. Overall, 29 patients were tested for BRAF mutations, out of which 12 were BRAF-mutated and treated with BRAF inhibitors, but its benefits have been limited due to acquired resistance. The combination of BRAF/MEK inhibitors has only recently been approved and no long-term follow-up for patients receiving this therapy is available for this study. Even though most of our patients were diagnosed with progressive disease and died during follow-up, significant progress has been made in managing advanced cutaneous melanomas, and OS has increased in recent years [27]. Increasing evidence argues in favor of sentinel lymph node biopsy (SLNB) for identifying high-risk patients that can benefit from systemic therapy [24,26,35,36,38]. Even though complete lymph node dissection does not influence PFS or OS [25], positive SNLB is associated with decreased PFS and OS and should be treated more aggressively [35,36,41]. Since positive SLNB in thick melanomas automatically increases staging from IIB/IIC to IIIC [29], such patients may be treated with adjuvant systemic therapy, including newer drugs such as immune-checkpoint inhibitors and/or BRAF/MEK inhibitors, which have been shown to improve PFS and OS [41,55,56,57,58,59]. Nevertheless, thick cutaneous melanomas have a poor prognosis even in cases of negative SLNB [34], and adjuvant therapy may be beneficial for all patients with melanomas > 4 mm regardless of SLNB status [60,61]. Consequently, the USA Food and Drug Administration (FDA) approved adjuvant immunotherapy for thick melanomas without nodal involvement (stages IIB/IIC) in 2021 [62]. For patients with metastatic disease, immune-checkpoint inhibitors and BRAF/MEK inhibitors remain valid therapeutic options, with superior overall response rates than conventional chemotherapy [63]. Additionally, stage III-IV may benefit from triple combination therapy with PD-1/PD-L1, BRAF, and MEK inhibitors. In this respect, a systematic review demonstrated that this combination improves PFS and OS but is associated with increased adverse effects [64].

On the contrary, the role of neoadjuvant therapy in advanced cutaneous melanomas is not established. Some authors reported promising results for neoadjuvant therapies such as immunomodulators [62,65] or BRAF/MEK inhibitors [66]. However, neoadjuvant therapy was also associated with considerable adverse effects [62,67], and a recent systematic review found no certain benefits of neoadjuvant therapy in advanced cutaneous melanomas [68].

Finally, several emerging therapies are currently being studied for advanced melanomas. So far, oncolytic viral therapy, engineered cytokines, anti-VEGF drugs, or T-cell agonists have shown promising results in clinical trials [63]. Further clinical trials are required to determine the best therapeutic approach for patients with thick melanomas, and addressing adverse prognostic factors is important for their management.

## 5. Conclusions

Thick melanomas are generally considered to have a poor outcome. Still, not all cases behave the same, and risk assessment is crucial to provide the best therapeutic approach for each patient. This study has addressed a large number of features that have been associated with prognosis in thick melanomas. We have shown that the only independent adverse prognostic factors are increasing Breslow depth, necrosis, microsatellites, and perineural invasion for both PFS and OS. Even for melanomas > 4 mm, the depth of invasion remains an important prognostic factor. Furthermore, this research is the first study to demonstrate a correlation between perineural invasion and aggressive behavior in thick melanomas and one of the few to address the influence of necrosis. Acknowledging adverse prognostic factors in thick cutaneous melanomas is particularly important in the era of personalized therapy since it may allow patients to benefit from the best therapeutic options.

## Figures and Tables

**Figure 1 biomedicines-11-02616-f001:**
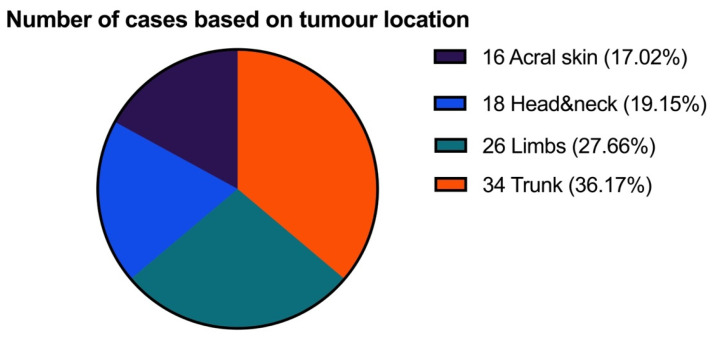
The distribution of primary tumor location.

**Figure 2 biomedicines-11-02616-f002:**
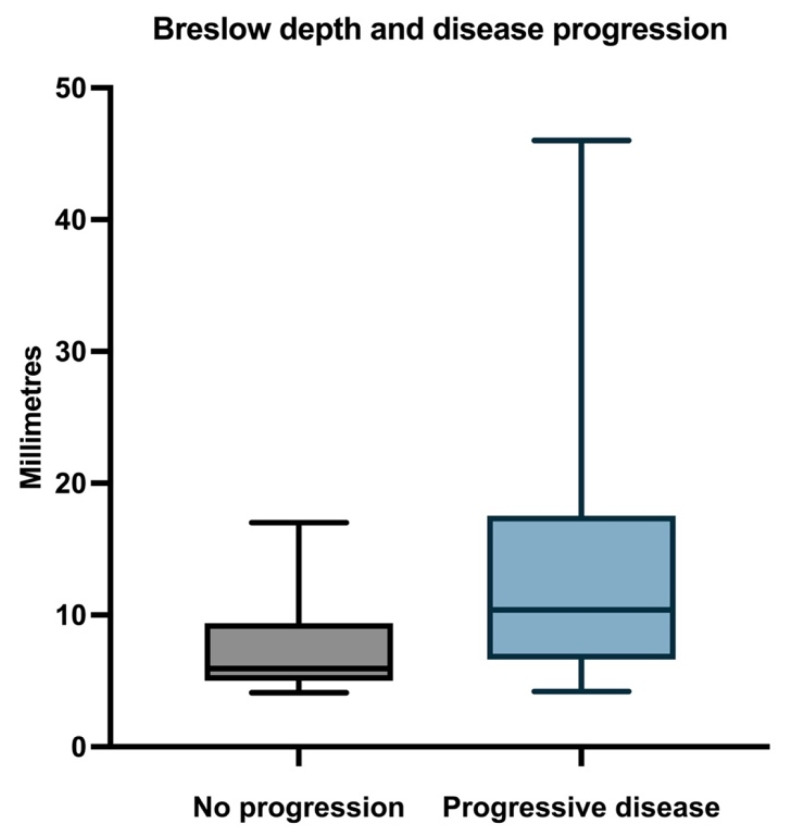
Median, minimum, and maximum Breslow depth in relation to PFS.

**Figure 3 biomedicines-11-02616-f003:**
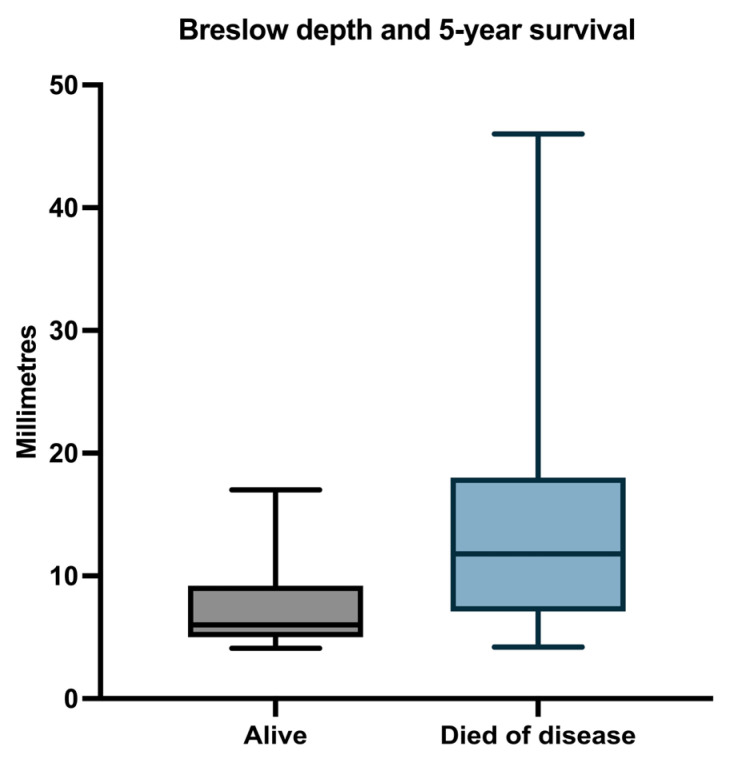
Median, minimum, and maximum Breslow depth in relation to OS.

**Figure 4 biomedicines-11-02616-f004:**
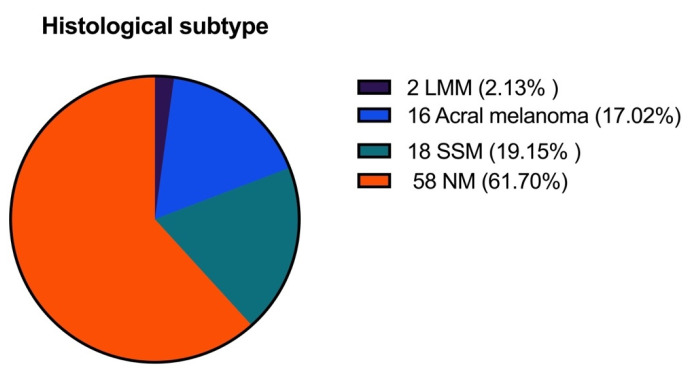
The distribution of melanoma histological subtypes in the whole study population.

**Figure 5 biomedicines-11-02616-f005:**
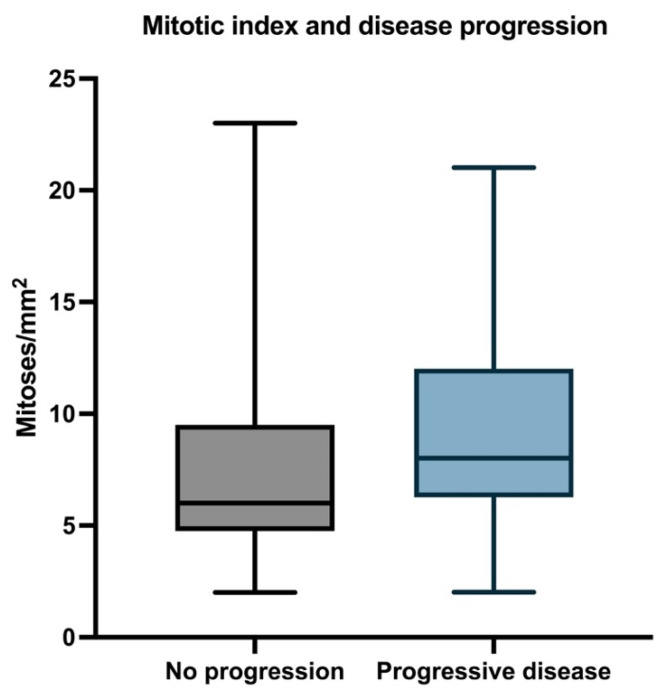
Median, minimum, and maximum mitotic count in relation to PFS.

**Figure 6 biomedicines-11-02616-f006:**
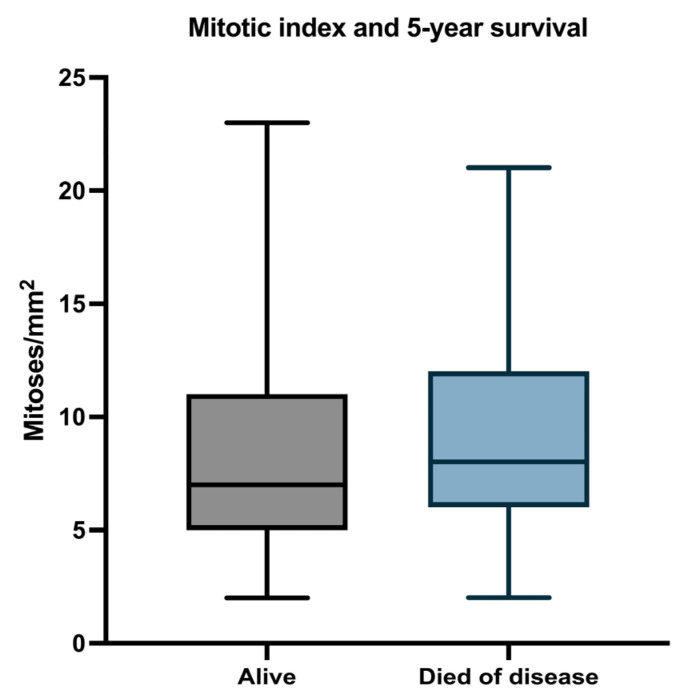
Median, minimum, and maximum mitotic count in relation to OS.

**Figure 7 biomedicines-11-02616-f007:**
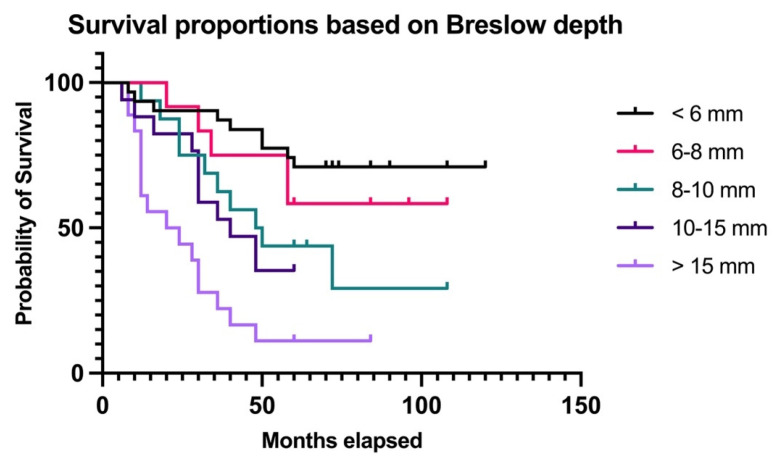
OS stratified by Breslow depth.

**Table 1 biomedicines-11-02616-t001:** Cox univariate analysis of clinical prognostic factors.

	PFS			OS		
	HR	95%CI	*p* Value	HR	95%CI	*p* Value
Age	1.010	0.99–1.03	0.3256	1.009	0.99–1.03	0.4012
Gender (Female)	0.49	0.28–0.83	0.0103	0.42	0.23–0.74	0.0036
Tumor location						
Head and neck	Ref.					
Trunk	0.89	0.43–1.92	0.7542	1.08	0.49–2.49	0.8569
Limbs	0.79	0.36–1.79	0.5563	0.87	0.37–2.13	0.7537
Acral skin	1.28	0.55–2.99	0.5688	1.49	0.62–3.7	0.3732

**Table 2 biomedicines-11-02616-t002:** Cox univariate analysis of the depth of invasion.

	PFS			OS		
	HR	95%CI	*p* Value	HR	95%CI	*p* Value
Breslow	1.08	1.05–1.11	<0.0001	1.08	1.05–1.11	<0.0001
Clark (V)	1.31	0.76–2.23	0.3207	1.35	0.76–2.35	0.3001

**Table 3 biomedicines-11-02616-t003:** Histological subtype in relation to PFS and OS.

	Nodular Melanoma	Superficial Spreading Melanoma	Acral Lentiginous Melanoma	Lentigo Maligna Melanoma	Total
No metastases	52.63% (n = 20)	31.58% (n = 12)	13.16% (n = 5)	2.63% (n = 1)	38
Metastases	67.86% (n = 38)	10.71% (n = 6)	19.64% (n = 11)	1.79% (n = 1)	56
Alive	53.49% (n = 23)	30.23% (n = 13)	11.63% (n = 5)	4.65% (n = 2)	43
Died	68.63% (n = 35)	9.8% (n = 5)	21.57% (n = 11)	0% (n = 0)	51

**Table 4 biomedicines-11-02616-t004:** Cox univariate analyses on the presence of ulceration, necrosis, regression, microsatellites, lymphovascular invasion, neurotropism and PFS.

	No Progression	Disease Progression	Total	HR (95% CI)	*p* Value
Ulceration	81.58% (n = 31)	94.64% (n = 53)	89.36% (n = 84)	2.44 (0.89–10.02)	*p* = 0.1338
Necrosis	18.42% (n = 7)	69.64% (n = 39)	48.93% (n = 46)	4.08 (2.33–7.44)	*p* < 0.0001
Regression	26.32% (n = 10)	12.5% (n = 7)	18.08% (n = 17)	0.53 (0.22–1.09)	*p* = 0.1171
Microsatellites	5.26% (n = 2)	32.14% (n = 18)	21.28% (n = 20)	3.17 (1.75–5.52)	*p* < 0.0001
Lymphovascular invasion	31.58% (n = 12)	37.5% (n = 21)	35.1% (n = 33)	1.39 (0.79–2.35)	*p* = 0.2457
Neurotropism	7.89% (n = 3)	37.5% (n = 21)	25.53% (n = 24)	3.37 (1.91–5.8)	*p* < 0.0001

**Table 5 biomedicines-11-02616-t005:** Cox univariate analyses on the presence of ulceration, necrosis, regression, microsatellites, lymphovascular invasion, neurotropism and OS.

	Alive	Died	Total	HR (95% CI)	*p*-Value
Ulceration	83.72% (n = 36)	94.12% (n = 48)	89.36% (n = 84)	2.12 (0.78–8.7)	*p* = 0.2086
Necrosis	23.26% (n = 10)	70.59% (n = 36)	48.93% (n = 46)	3.96 (2.2–7.49)	*p* < 0.0001
Regression	23.26% (n = 10)	13.73% (n = 7)	18.08% (n = 17)	0.58 (0.24–1.21)	*p* = 0.1835
Microsatellites	4.65% (n = 2)	35.29% (n = 18)	21.28% (n = 20)	3.6 (1.98–6.38)	*p* < 0.0001
Lymphovascular invasion	24.53% (n = 13)	39.22% (n = 20)	35.1% (n = 33)	1.53 (0.86–2.67)	*p* = 0.1293
Neurotropism	9.09% (n = 3)	41.18% (n = 21)	25.53% (n = 24)	35.61 (8.94–241.9)	*p* < 0.0001

**Table 6 biomedicines-11-02616-t006:** Cox multivariate analysis for PFS.

Characteristic	HR	CI 95%	*p* Value
Necrosis	2.27	1.21–4.42	0.0126
Microsatellites	2.59	1.33–5.8	0.0069
Neurotropism	2.45	1.21–4.93	0.0117
Breslow depth	1.04	1.005–1.07	0.0205
Gender (female)	0.76	0.34–1.42	0.3943
SSM	0.63	0.23–1.49	0.3249
Mitosis	1.02	0.96–1.08	0.4288

**Table 7 biomedicines-11-02616-t007:** Cox multivariate analysis for OS.

Characteristic	HR	CI 95%	*p* Value
Necrosis	2.19	1.14–4.34	0.0203
Microsatellites	2.84	1.42–5.5	0.0024
Neurotropism	2.19	1.09–4.32	0.0249
Breslow depth	1.038	1.005–1.07	0.0177
Gender (female)	0.61	0.32–1.16	0.1346
SSM	0.56	0.19–1.33	0.2339

## Data Availability

All the data processed in this article are part of the research for a doctoral thesis, which is archived in the pathology department at the University Hospital of Bucharest where the interventions were performed. The original data are available upon reasonable request.

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
