# Peer review of "The Impact of Clinical and Histopathological Factors on Disease Progression and Survival in Thick Cutaneous Melanomas"

_biomedicines, 2023, doi:10.3390/biomedicines11102616_

Round 1

Reviewer 1 Report

dear authors, I appreciated your manuscript that involved a great case series of thick melanomas from a single istitution.

The main criticism is that you seem not to have taken into account the eventual adiuvant treatment for survival during follow-up. This aspect should be better explained (I see only few lines in the discussion).

Furthermore, among the pathologic markers, the study of BRAF could be added.

Another criticism from my point of view is a missing dermoscopic characterizations of melanomas. I understand that your study has been performed on histological reports but for dermatologists the diagnostic possibility of predict the severity of melanoma could be really important. Some attempts can be found in the literature (for example Niforou A et al, JEADV 2021 - Sgouros D et al JEADV 2020). This could be added in a limitation paragraph. Also anamnesis and clinical information could be important in the survival prediction.

- in the figure 3 and 5 there is a mispelling: "disease" should be correct

Author Response

Response to Reviewer 1 Comments

We want to thank the reviewer for their observations. In response to your thoughtful feedback, we have carefully considered each of your points and have addressed them in a point-by-point manner below. The revised paper contains changes according to comments. As indicated, we highlighted what was added to make it easier to track our changes. In addition, the text was subjected again to grammar and spelling checking -all modifications are highlighted as indicated.

Point 1: The main criticism is that you seem not to have taken into account the eventual adiuvant treatment for survival during follow-up. This aspect should be better explained (I see only few lines in the discussion).

Response 1: We appreciate your valuable input. We have added more information about the treatment the patients received during follow-up. We have not explored this aspect in much detail since, in most cases, the patients in this study received therapies that are no longer the golden standard of care. This may however explain the low survival rates in our group.

Point 2: Furthermore, among the pathologic markers, the study of BRAF could be added.

Response 2: We are grateful for this suggestion and therefore mentioned the cases that were genetically tested for BRAF mutations. Unfortunately, this number is rather small as at the time our study began, BRAF inhibitor therapy was not available in Romania. We agree that retrospective immunohistochemical analysis of BRAF would offer some interesting results and plan to evaluate this marker in future research.

Point 3: Another criticism from my point of view is a missing dermoscopic characterizations of melanomas. I understand that your study has been performed on histological reports but for dermatologists the diagnostic possibility of predict the severity of melanoma could be really important. Some attempts can be found in the literature (for example Niforou A et al, JEADV 2021 - Sgouros D et al JEADV 2020). This could be added in a limitation paragraph. Also anamnesis and clinical information could be important in the survival prediction.

Response 3: We found your suggestion very interesting and as a consequence, we added a new paragraph discussing the references you mentioned and their relevance to our study.

Point 4: - in the figure 3 and 5 there is a mispelling: "disease" should be correct

Response 4: We thank you for all these remarks and apologize for the typos. We have modified all of them accordingly.

Reviewer 2 Report

Dear Editor and Authors,

Thank you for your invitation to review this manuscript titled “The Impact of Clinical and Histopathological Factors on Disease Progression and Survival in Thick Cutaneous Melanoma” by Dr. Țăpoi and her colleagues from Romania.

This is a well written and well presented paper which describes the factors that can influence outcomes in patients with Thick Cutaneous Melanoma! The subject is interesting and has clinical significance.

I have only some minor comments to make:

1. How were variables and data gathered for the study patients? Was there a retrospective chart review performed or was there a dedicated departmental database which was mined for data? What percentage of missing data was there?

2. How was the TNM 8th edition used for stage classification considering the study period is from 2012 which is well before it came out? Were patients re-examined and re-classified?

3. The histopathological analysis section is not really needed (we don’t need to know you looked the 10% neutrally buffered formalin, paraffin embedded, sectioned, and stained with Hematoxylin-Eosin sections under a Nikon E200 light microscope)!! Please, only focus on the pertinent – methodology related information (two pathologist were used with a third for a tie break!).

4. Line 179 in results. Is it SSM or SMM?

5. The number of LMM is quite small (2 patients) so I would quantify the statement that no patient of LMM died as the sample is too small for reliable results!

6. It is interesting to note that resection margins were not correlated to OS. Why do the author this is?

In conclusion, I found this study quite well done and thorough! I only have a few comments and queries which when answered I will probably be happy to recommend the publication of this work. Kind regards to all.

Minor editing required. OK

Author Response

Response to Reviewer 2 Comments

We want to thank the reviewer for their observations. In response to your thoughtful feedback, we have carefully considered each of your points and have addressed them in a point-by-point manner below. The revised paper contains changes according to comments. As indicated, we highlighted what was added to make it easier to track our changes. In addition, the text was subjected again to grammar and spelling checking -all modifications are highlighted as indicated.

Point 1: How were variables and data gathered for the study patients? Was there a retrospective chart review performed or was there a dedicated departmental database which was mined for data? What percentage of missing data was there?

Response 1: We appreciate your valuable input. We have performed a retrospective chart review and we have added more information about how the data was collected and about missing data.

Point 2: How was the TNM 8th edition used for stage classification considering the study period is from 2012 which is well before it came out? Were patients re-examined and re-classified?

Response 2: Thank you for this observation. We have now mentioned that at the before beginning the work on this manuscript we re-evaluated all the cases and classified them according to the latest WHO and AJCC editions.

Point 3: The histopathological analysis section is not really needed (we don’t need to know you looked the 10% neutrally buffered formalin, paraffin embedded, sectioned, and stained with Hematoxylin-Eosin sections under a Nikon E200 light microscope)!! Please, only focus on the pertinent – methodology related information (two pathologist were used with a third for a tie break!)

Response 3: We appreciate these recommendations and have modified the paragraph accordingly.

Point 4: Line 179 in results. Is it SSM or SMM?

Response 4: We apologize for the typos. It should be SSM, and we have modified the text.

Point 5: The number of LMM is quite small (2 patients) so I would quantify the statement that no patient of LMM died as the sample is too small for reliable results!

Response 5: Thank you very much for your remark. We added this statement in the discussion section.

Point 6: It is interesting to note that resection margins were not correlated to OS. Why do the author this is?

Response 6: We appreciate this interesting observation and we have made some comments on this matter in the discussion section. We believe that resection margins were not correlated with OS as they were on average only slightly larger in patients who survived than in patients who died. We believe that resection margins could influence PFS and OS if they were larger than 2 cm, but this was generally not the case in our study.